# C1q Binds to CD4+ T Cells and Inhibits the Release of Pro-Inflammatory Cytokines: Role in the Pathogenesis of Systemic Lupus Erythematosus

**DOI:** 10.3390/ijms26104468

**Published:** 2025-05-08

**Authors:** Arushi Dogra, Anne G. Savitt, Berhane Ghebrehiwet

**Affiliations:** 1Division of Rheumatology, Allergy and Clinical Immunology, Stony Brook University, Stony Brook, NY 11794, USA; 2Department of Microbiology and Immunology, Renaissance School of Medicine, Stony Brook University, Stony Brook, NY 11794, USA

**Keywords:** C1q, systemic lupus erythematosus, CD4+, T cells, proliferation, cytokines, inflammation

## Abstract

The association between C1q deficiency and the development of Systemic Lupus Erythematosus (SLE) is well established. Several studies have shown that deficiency in C1q is associated with failed apoptotic cleanup, leading to SLE progression. However, the magnitude of this correlation indicates that C1q may play a much more complex role in the development of lupus. This study provides further insight into the pathogenesis of SLE by investigating the consequences of the interaction between C1q and CD4+ T-cells in the breakdown of self-tolerance. Since the C1q/C1q receptor interaction is postulated to play a role, we first confirmed the presence of surface-expressed C1q and C1q receptors on CD4+ T-cells. Then, cell proliferation assays were performed in the presence and absence of purified C1q, gC1qR, and cC1qR. The supernatants of these cultures were used to determine the levels of immunoregulatory cytokines released. Our data confirm that increasing concentrations of C1q and gC1qR significantly inhibited cell proliferation. Furthermore, the CD4+ cells treated with either C1q or gC1qR secreted reduced inflammatory cytokines, such as IL-6 and TNF-alpha, compared to the untreated controls, suggesting that C1q deficiency facilitates the uncontrolled secretion of these critical cytokines, thus contributing to SLE. Although the role of pro-inflammatory cytokines in the induction of SLE is well documented, the mechanism by which C1q contributes to the disease is still a study in progress. Our data demonstrate that the interaction between C1q and its receptors on CD4+ T cells plays a critical role in the suppression of pro-inflammatory cytokines that cause tissue injury in SLE. Therefore, the C1q-C1qR axis may provide a rationally sound target for the design of novel therapeutic approaches for SLE treatment.

## 1. Introduction

Systemic Lupus Erythematosus (SLE) is a chronic autoimmune disease that causes connective tissue inflammation, often leading to muscle pain, weakness, and rashes, as well as complications such as nephritis, pericarditis, or peripheral neuropathy. While much progress has been made in understanding the pathogenesis of SLE over the past few decades, there are still many unanswered questions with regard to the factors that lead to its development. Nonetheless, numerous genetic and environmental factors are postulated to play a major role [1,2,3,4,5,6,7,8]. One condition with a strong correlation to SLE is deficiency in the complement system molecule C1q [3,5,6,7,8]. Although it is a rare multifactorial disease characterized by inflammation in several organ systems, there is abundant clinical evidence showing that homozygous deficiency in any of the classical complement pathway proteins—C1q, C1r, C1s, C4, and C2—predisposes an individual to SLE and other autoimmune diseases [3]. These proteins function as opsonins and trigger membrane attack complex (MAC) formation through interactions with corresponding cell-surface receptors. Among them, C1q takes center stage in significance, as the recognition component of the C1 complex and the pathway initiator. C1q is a glycoprotein composed of 18 polypeptide chains organized into six globular heads and a collagenous tail. The head and tail domains of the C1q bind to the gC1qR and cC1qR cell-surface receptors, respectively, to function [8]. Hereditary homozygous deficiency due to mutations in the C1q gene—predominantly a mutation in the A-chain, in which the C to T transition in codon 186 of exon 2 results in Gln-to-Stop (Q186X) substitution—is a powerful susceptibility factor for the development of not only SLE, but also other diseases such as angioedema and rheumatoid arthritis [5,6]. In fact, more than 90% of individuals with an inherited C1q deficiency go on to develop SLE. This deficiency is postulated to contribute to SLE development due to the consequent failure of the adequate clearance of apoptotic cells and immune complexes [7,8,9,10,11,12,13,14,15]. However, due to the strong correlation between C1q deficiency and SLE and the existence of alternative and redundant immune mechanisms to account for apoptotic cell clearance, we postulate that C1q and its corresponding receptors play a much more prevalent role in lupus immunoregulation and autoimmunity [16].

In addition to C1q deficiency, lupus is also hallmarked by an enhanced immune response in CD4+ T-cells. This contributes to a loss of self-tolerance in SLE patients, or the failure of the immune system to not react against one’s own antigens [3,17]. CD4+ T-cells, also known as helper T-cells (Th), are thymus-derived lymphocytes that are an essential part of the adaptive immune response. They target specific pathogenic antigens recognized as non-self by the T-cell receptor (TCR) by binding to major histocompatibility complex (MHC) Class II molecules on the foreign antigen-presenting cells. After being activated, naive helper T-cells differentiate into several effector cell types with diverse functions, including the activation of other innate and adaptive immune cells, the creation of memory and plasma B-cells, and the inhibition of exaggerated immune reactions when appropriate [18,19]. They do this through cytokine secretion, which is triggered by surface receptor–ligand binding. These pro- or anti-inflammatory cytokines then continue to cascade through interactions with T-cell receptors or the receptors of other surrounding immune and somatic cells [19].

Autoimmune disorders such as lupus develop due to the loss of tolerance to self-antigens. This is driven by the erroneous activation of autoreactive T-cells or the loss of function in regulator T-cells (Treg), a subset of the CD4+ T-cells [20]. In SLE, anti-self Th cells breakdown immunity. This is evidenced by the exaggerated inflammatory response in lupus, involving the elevated proliferation of CD4+ T-cells, excessive activation of B-cells and dendritic cells, and altered cytokine secretion at inflammation sites [21]. Additionally, previous studies have shown that culturing CD4+ T-cells with C1q inhibits proliferation in a dose-dependent manner, suggesting that a lack of C1q-mediated T-cell immunosuppression could contribute to SLE pathogenesis [21,22]. Further studies on the role of CD4+ T-cells in the development of SLE and autoimmunity could, therefore, lead to the identification of new therapeutic targets for SLE.

Most functions of the immune system, including the activation of immune cells and the maintenance of self-tolerance, are ultimately regulated by the release of specific pro-inflammatory and anti-inflammatory cytokines, which are small molecules that mediate both the innate and adaptive immune responses [23]. The major cytokines secreted by CD4+ T-cells and considered essential for normative immune function are interferon gamma (IFN-γ), tumor necrosis factor alpha (TNF-α), transforming growth factor beta (TGF-β), and several interleukins (ILs) [18]. Of these, those also associated with the development of SLE are IL-2, IL-4, IL-6, IL-10, IL-17, IFN-γ, TNF-α, and TGF-β [23]. While IL-2, IL-6, IFN-γ, and TNF-α are thought to be pro-inflammatory cytokines [24], IL-4 is known to have anti-inflammatory functions. Treg cells secrete IL-10 and TGF-β to suppress autoreactive helper cell activation, preventing autoimmunity [19,24]. Additionally, interestingly, while IFN-γ is centrally recognized for its pro-inflammatory effect and phagocytic activation, some studies have also shown that IFN-γ suppresses the inflammatory response, promoting Treg differentiation and apoptotic mediator expression [18,25,26]. Therefore, it is believed to have both pro-inflammatory and anti-inflammatory effects [23].

In the present work, we investigated IL-6 and TNF-α as pro-inflammatory cytokines and IL-10 and IFN-γ as anti-inflammatory cytokines, all of which are elevated in serum samples of SLE patients and associated with disease activity [23]. The first of these, IL-6, is secreted by macrophages in response to the binding of pathogens to cell-surface receptors, such as Toll-like receptors. This induces the cellular transduction pathways leading to TNF-α and further IL-6 production, establishing a positive feedback loop of immune stimulation. Some known functions of IL-6 are acute phase inflammatory protein synthesis, neutrophil production in the bone marrow, and Treg cell suppression, thereby promoting immunogenicity and decreasing tolerance [27]. Mouse model experiments also found that IL-6 can specifically exacerbate T-cell driven immunity by inducing IL-17 and promoting T-cell proliferation [28]. TNF-α works in conjunction with IL-6 in many of these pathways, including the acute phase response, and can trigger local inflammatory symptoms such as heat, swelling, etc. Additionally, TNF-α assists with neutrophil migration and induced phagocytosis [29]. In contrast, IL-10 inhibits the synthesis of pro-inflammatory cytokines, including IL-2, IL-3, TNF-α, and granulocyte–macrophage colony-stimulating factor (GM-CSF), a hematopoietic glycoprotein produced by macrophages and Th1 cells assisting with white blood cell production. IL-10 has also been shown to suppress immune cell capacity to present antigens and trigger an immune response [30]. As mentioned earlier, IFN- γ has pleiotropic functions. Under normal immune conditions, it serves to activate macrophages and induces the expression of MHC Class II proteins [31,32,33]. However, it has also been shown to contradictorily ameliorate autoimmune disorders like SLE by suppressing the inflammatory response, increasing the expression of apoptotic mediators to cause cell death, and suppressing the differentiation of Th17 cells and osteoclasts [25].

With both C1q deficiency and irregular CD4 + T-cell signaling playing major roles in SLE development, we investigated the interaction of these two factors to gain new insight into lupus progression. These experiments aim to re-affirm C1q’s inhibition of T-cell proliferation and explore the mechanism of this function via the relationship between C1q and its receptors, gC1qR and cC1qR, and the secretion of the identified central cytokines. The results of these studies will not only give better insight into the role of C1q in the development of SLE, but could also be used to design novel therapeutic options for lupus treatment.

## 2. Results

### 2.1. Cell-Surface Expression of C1q, gC1qR, and cC1qR on Molt-4 Cells

Although the presence of cell-surface gC1qR and cC1qR has been shown before [8], the surface expression of C1q and its receptors on the Molt-4 cells, which are used in these studies as a surrogate for CD4+ T-cells, needed to be verified using an indirect ELISA in order to confirm the suitability of the cell line for serving as an appropriately representative substitution. The analysis of the indirect ELISA on the microtiter plate-fixed cells confirmed the presence of surface-expressed C1q as well as both receptors, gC1qR and cC1qR (Figure 1), affirming the choice of cell line. The surface proteins were detected using specific antibodies to C1q, gC1qR, and cC1qR (*p*-value > 0.001, n = 4).

### 2.2. Effect of C1q on Proliferation of Molt-4 T-Cells

To assess the effect of C1q on the Molt-4 T-cells, a cell proliferation assay was conducted using the trypan blue exclusion assay, in which the numbers of live and dead cells were analyzed in samples taken from cultures of the cells incubated (0–96 h, 37 °C) with concentrations of C1q ranging from 0 to 20 μg/mL. Since the concentration of C1q in plasma is approximately 90–110 µg/mL, the highest concentration used in the experiment was still 5× less than the normal concentration of C1q. The assay was run in duplicates and repeated three times, with the viability assay performed at 24 h intervals. As shown in Figure 2a, the increasing concentration of C1q in the T-cell cultures inhibited cell proliferation, with the culture with 20 μg/mL of C1q having less than 50% of the number of cells in the control after 96 h. The relatively small bars of standard error illustrate the significance in the differences between the counts for each experimental group.

In order to ensure that this difference in the live cells was due to effects on cell proliferation, rather than cell death, the presence of dead cells was also counted at 24 h intervals, and showed that the numbers of dead cells with each concentration of C1q remained similar in all samples tested (Figure 2b). After 96 h, a comparison of the cultures with and without C1q showed insignificant numbers of dead cells, indicating that C1q is not toxic.

The effect of C1q on Molt-4 cell proliferation was also visually compared through representative pictures taken of the cell cultures at 24 h intervals, at the same times when the viability studies were done. The difference between the untreated control and the culture with 20 μg/mL of C1q was apparent after 96 h, with a clear difference between the confluence of cells in the control culture and the diminished numbers in the culture with 20 μg/mL of C1q (Figure 3).

### 2.3. Effects of Increasing Concentrations of gC1qR on Proliferation of Molt-4 T-Cells

Since CD4+ T-cells have been shown to express surface C1q, we investigated to see how the binding of gC1qR and cC1qR to membrane-associated C1q impacted the survival or proliferation of the Molt-4 T-cells. To quantitatively measure the effects of gC1qR, cell proliferation assays were conducted by assessing the numbers of live and dead cells in cultures of CD4+ T-cells after incubation with various concentrations of gC1qR over a period of 96 h. The cells were counted at 24 h intervals, and each assay was performed in duplicate and repeated three times (n = 3). As shown in Figure 4a, the increasing concentrations of gC1qR suppressed T-cell proliferation, with the maximal effect seen at 20 μg/mL of C1q. Analysis of the dead cell count also revealed that the number of dead cells was insignificant, since the cell count with each concentration of gC1qR remained unchanged, indicating that gC1qR is not cytotoxic, but rather suppressive (Figure 4b). At the 96 h mark, the control sample contained 157.500 ± 9.5742 × 10^3^ dead cells, while the culture incubated with 20 μg/mL of gC1qR contained 165.000 ± 12.9099 × 10^3^ dead cells. Overall, considering the impact of the treatment on both the live and dead cell counts, the effects of C1q and gC1qR on T-cell proliferation were extremely similar.

The effects of the increasing gC1qR concentrations on CD4+ T-cell proliferation were also detected qualitatively. The difference between the control and the 20 μg/mL gC1qR culture was most qualitatively apparent after 96 h, with a clear difference in the density of cells between the control culture and that in the 20 μg/mL gC1qR culture (Figure 5).

### 2.4. Effects of Increasing Concentrations of cC1qR on Proliferation of CD4+ T-Cells

Similarly, the impact of cC1qR on the T-cell proliferation rate was also studied. The concentrations of cC1qR tested were 0 μg/mL, 5 μg/mL, 10 μg/mL, and 20 μg/mL. The cells were counted at 24 h intervals as described earlier. Contrary to what was observed with C1q and gC1qR, the increasing concentrations of cC1qR in the T-cell cultures had no significant impact on the numbers of live cells in the cultures, with the control culture having 130.000 ± 4.0825 × 10^4^ live cells at 96 h and the culture incubated with 20 μg/mL of cC1qR having 134.05 ± 5.0580 × 10^4^ live cells (Figure 6a). Both of these values are within one standard deviation of each other, illustrating the relative insignificance of their difference.

In order to determine whether the incubation with cC1qR impacted the viability of the cells, the dead cells were counted at 24 h intervals as well. Analysis of the dead cell counts at 24 h intervals revealed that the number of dead cells with each concentration of cC1qR remained extremely similar (Figure 6b). At each 24 h point, all four samples had an insignificant number of dead cells. At the 96 h mark, the control sample contained 156.250 ± 14.4539 × 10^3^ dead cells, while the culture incubated with 20 μg/mL of gC1qR contained 146.500 ± 9.8107 × 10^3^ dead cells. With a difference of only 9.75 thousand cells in their means, these cell counts are both less than one standard deviation apart, which is not significant.

The effects of the increasing cC1qR concentrations on CD4+ T-cell proliferation were also detected qualitatively, through pictures taken at 24 h intervals. The similarity between the control and the 20 μg/mL cC1qR culture was qualitatively similar at each time point (Figure 7).

### 2.5. The Role of C1q, gC1qR, and cC1qR in the Release of Key Regulatory Cytokines from T-Cells

Of the cytokines secreted by CD4+ T-cells and involved in the development of SLE, the most significant are IL- 6, TNF-α, IL-10, and IFN-γ. Of these, IL-6, TNF-α, and IFN-γ are immunostimulants, promoting the immune response to pathogens, while IL-10 is an immunosuppressant. However, in the context of autoimmunity, multiple studies have shown that IFN-γ can also act as an anti-inflammatory cytokine. Interestingly, IL-6, TNF-α, IL-10, and IFN-γ are all often found at higher-than-normal concentrations in SLE patients, indicating that their anti-inflammatory functionality may also contribute to disease development [23].

To explore the regulatory functions of C1q and its receptors on the immune responses of CD4+ T-cells, direct ELISAs were conducted on supernatant samples of Molt-4 cultures grown with 10 μg/mL of C1q, gC1qR, or cC1qR, and the levels of secreted IL-6, TNF-α, IL-10, and IFN-γ were measured and compared to untreated T-cells. The ELISA was run in duplicate and repeated four times. As shown in Figure 8a,b, the levels of both IL-6 and TNF-α were slightly downregulated in the T-cells treated with C1q and gC1qR. The significance of this was shown by *p*-values that were less than 0.05, or less than 0.01 for IL-6 expression. In contrast, neither the IL-10 nor IFN-γ levels were significantly altered by any of the treatments with C1q, gC1qR, or cC1qR (Figure 9a,b). Although there were slight differences in the values of the relative absorbance between treatments, these differences were not large enough to warrant significance. All *p*-values were greater than 0.05.

The results pertaining to the effects of C1q, gC1qR, and cC1qR in the cell cultures on the release of pro-inflammatory and anti-inflammatory cytokines by the CD4+ T-cells are summarized in Table 1.

## 3. Discussion

The role of the complement system, and the role of C1q in particular, in the pathogenesis of SLE is very well established [1,2,3,4,5,6,7,8]. However, previous research has singularly focused on proving the postulate that deficiency in C1q leads to the inadequate removal of apoptotic cells, thereby contributing to autoimmune responses and lupus progression. Although C1q may indeed play a role in the removal of apoptotic debris, there are other well-orchestrated mechanisms that would ensure that apoptotic cleanup is duly regulated, even in the absence of C1q [12]. In addition, although C1q expression on the surface of a wide range of cell types has been demonstrated [12], the functions of cell-surface C1q and of cells that express both C1q and the gC1qR receptor are still incompletely understood. This study therefore focused on examining the role of cell-surface-expressed C1q and its function in SLE in order to understand why a deficiency in C1q predisposes one to develop autoimmune diseases such as lupus. Since cytokines play an essential role in regulating inflammation, we investigated whether the influence of C1q on the secretion and suppression of specific cytokines could be, in part, responsible for the loss of self-tolerance that largely contributes to the development and progression of SLE.

Our studies show that CD4+ T-cells expressed cell-surface C1q, in addition to the receptors for its globular and collagen-like regions, gC1qR and cC1qR, respectively. Therefore, under the assumption that the logic of “structure-serves-function” holds true, this finding likely signifies that C1q and its receptors play essential roles in the regulation of the T-cell immune response. Employing a cell proliferation assay, we show here that the culturing of T-cells with C1q did, in fact, significantly inhibit the proliferation of the T-cells, which is consistent with our previously published observations [22]. Since the live cell counts decreased while the dead cell counts remained the same, C1q appears to inhibit the proliferation of these cells without damaging them. Furthermore, this method was used to show that treating CD4+ T-cells with gC1qR also inhibited proliferation in a similar fashion. In contrast, cC1qR did not significantly impact T-cell proliferation. This is postulated to be a result of the fact that cC1qR binds predominantly to a region in the collagen tail of the C1q A-chain, through which the molecule is anchored to the cell membrane, thereby making the cC1qR site unavailable for binding. Alternatively, this could also signify that the binding of cC1qR to surface C1q in T-cells may have a heretofore unidentified function that relates to another means of lupus development.

These findings raise an interesting proposition for the mechanism by which T-cells independently regulate population growth intercellularly. If T-cells have both gC1qR and the exposed globular heads of C1q on their surfaces, gC1qR may be released into the pericellular milieu as the population density increases and cell crowding occurs, which we have previously demonstrated [22]. The surface C1q of one T-cell will then easily come into contact with the globular receptor (gC1qR) of another, and this binding could trigger an anti-proliferative pathway and control the T-cell population size. More importantly, it also suggests that genetic C1q deficiency could cause a loss of this self-regulation, as the deficient T-cells which lack soluble and/or cell-surface-expressed C1q would proliferate uncontrollably, and the inhibitory effects of surface C1q on other T-cells would decrease. Overactivity and loss of tolerance could, in turn, easily lead to the central symptoms seen in SLE. Furthermore, the presence of abnormally high levels of C1q would be expected to diminish the proliferation of CD4+ T-cells, and thus hinder a healthy immune response. Similarly, a C1q molecule displayed on any malignant cell could suppress the proliferative potential of a T-cell by binding to gC1qR in a manner akin to PD-L1 and PD-1 interactions.

To better understand the role of C1q in T-cell proliferation, we also investigated its relationship with the secretion of the key cytokines IL-6, TNF-α, IL-10, and IFN-γ, all of which play a role in the development of SLE and are upregulated in diseased serum samples. Of these cytokines, only the levels of IL-6 and TNF-α, which are known to contribute to the induction of a hyperactive immune system, were significantly inhibited by C1q and gC1qR. None of the cytokine levels were affected by cC1qR. This observation mirrors the results of previous experiments, which demonstrated that the co-culturing of T cells with C1q or gC1qR limits T cell proliferation [22]. The lack of effect observed with cC1qR further reinforces the suggestion that cC1q does not contribute to lupus development through the regulation of T cell proliferation and cytokine production, though it may potentially do so via other indirect pathways.

In summary, the results obtained in these studies seem to suggest that C1q deficiency is intimately linked to the overproduction of the potent pro-inflammatory cytokines IL-6 and TNF-α. Previous studies have demonstrated that IL-6 regulates and promotes CD4+ T cell proliferation, thereby creating an exaggerated immune response and thus leading to autoimmunity and the development of SLE. Further investigation into the roles of C1q and gC1qR on the surface of CD4+ T cells could not only provide new insight into the roles of these molecules in the immune activation leading to SLE, but may also help in the design of new therapeutic treatment options in a manner similar to treatments of angioedema with recombinant C1.

## 4. Materials and Methods

### 4.1. Chemicals and General Reagents

The following reagents and chemicals were purchased or obtained from the sources indicated: RPMI 1640, 100× Penicillin/Streptomycin, and trypsin/EDTA (Invitrogen, Carlsbad, CA, USA); heat-inactivated FBS (Hyclone, Logan, UT, USA); Dulbecco’s PBS (D-PBS) without calcium and magnesium (Mediatech Inc, Manassas, VA, USA); alkaline phosphatase (AP)-conjugated Streptavidin (Invitrogen, Carlsbad, CA, USA); p-nitrophenyl phosphate (pNPP) (Pierce, Rockford, IL, USA); and BSA (Thermo Fisher, Waltham, MA, USA).

### 4.2. Proteins and Antibodies

Rabbit antibodies to C1q and recombinant human gC1qR/p33, as well as human cC1qR/calreticulin, were made and purified in our laboratory and have been described in previous publications [32,33]. Purified human C1q was purchased from Quidel (San Diego, CA, USA). The receptors of the globular heads and collagen stalks of C1q, gC1qR, and cC1qR were purified from human cell lines, as described in previous publications [34]. Cytokine antibodies, such as rat anti-IL-6, anti-IL-10, anti-IFN-γ, or anti-TNF-α, were obtained from Thermo Fisher (Waltham, MA, USA).

### 4.3. Cultured Cells

The Molt-4 cell line was originally developed from cells taken from a 19-year-old male patient with T cell lymphoblastic acute cell leukemia in relapse (ATCC CRL-1582). Molt-4 CD4+ T cells can be utilized as proxies for SLE CD4+ T cells in mechanistic and molecular signaling experiments such as these. They provide an advantage in experimental convenience due to their indefinite proliferation and easier accessibility than primary SLE patient samples. Furthermore, we have successfully used these cells previously to show that the incubation of these cells with C1q induces an anti-proliferative response [22]. The cells were incubated in a 37 °C incubator with 5% CO_2_ and grown in RPMI 1640 medium with 10% FBS and 1% Penicillin/Streptomycin antibiotic solution (Pen-Strep).

The cell suspension was incubated in 10 mL of media in a T25 Falcon flask. After 48 h intervals, 100 μL of cells were taken out, to which 10 μL of Trypan blue dye was added, and the cells were evaluated for viability by counting under a microscope using a hematocytometer.

### 4.4. Enzyme-Linked Immunosorbent Assays (ELISAs)

Both indirect and direct whole-cell ELISAs were conducted in clear, 96-well, flat-bottomed, tissue culture-treated Falcon plates. For the indirect ELISAs, the plates were first coated with poly-L-lysine, and 10^5^ Molt-4 cells per well suspended in 100 μL of PBS were then added. The plates were centrifuged for 5 min at 1500 rpm and incubated with 100 μL of 0.5% glutaraldehyde in cold PBS for 30 min. The wells were washed with PBS and the unreacted sites blocked with 0.1% BSA in PBS. After washing, an indirect ELISA was conducted with primary Rabbit anti-C1q, anti-gC1qR, or anti-cC1qR, followed by reaction with AP-Donkey-anti-Rabbit. Then, 100 μL of diluted pNPP was added for color development, and the relative absorbance of the wells was measured at 405 nm using an ELISA plate reader.

For the direct ELISAs, 10^6^ Molt-4 cells were first cultured (37 °C for 36 h) in 1 mL of RPMI 1640 medium with 10% FBS and 1% Pen-Strep in duplicate wells of a 24-well, flat-bottomed, tissue culture-treated plate in the presence or absence of 10 μg/mL of either C1q, gC1qR, or cC1qR. After incubation, the samples were centrifuged for 5 min at 1100 rpm, and 100 μL of supernatant was collected. The wells of a 96-well plate were then coated with the supernatant, and after incubation (1 h, 37 °C), the wells were blocked with 300 μL of TBS-BSA. Then, biotinylated rat anti-IL-6, anti-IL-10, anti-IFN-γ, or anti-TNF-α were added and incubated for 60 min at 37 °C, followed by additional incubation with AP-conjugated Streptavidin. Finally, the color development was visualized by the addition of 100 μL of diluted pNPP, and the relative absorbance of the wells was measured at 405 nm using a plate reader.

### 4.5. Cell Proliferation Assays

The Molt-4 cells used in the cell proliferation assays were cultured as described above at 37 °C with 5% CO_2_ in RPMI 1640 medium, 10% FBS, and 1% Pen-Strep. The cells were plated in clear 24-well flat-bottomed Falcon plates, with each well containing 1 mL of 1 × 10^6^ cells/mL. The cells were then treated with increasing concentrations (0 μg/mL, 5 μg/mL, 10 μg/mL, and 20 μg/mL) of purified C1q, gC1qR, or cC1qR.

The assay was run for a total of 96 h, allowing the cells to grow in the treated media at 37 °C with 5% carbon dioxide (CO_2_). At 24 h intervals, data were collected from an undisturbed column of wells with two samples of each treatment, and the cells were photographed. Then, 100 μL of each sample was taken out, and the cell viability was assessed by the addition of 10 μL of Trypan blue dye as described above.

### 4.6. Statistical Analysis

Statistical analyses were calculated using GraphPad Prism version 8.0 (GraphPad Software, San Diego, CA, USA) and Excel (Microsoft, Redmond, Washington, NA, USA). Nonparametric ANOVA tests were applied to compare the control and experimental groups throughout. Differences were considered significant for *p* < 0.05 (n = separate experiments performed in duplicates).

## Figures and Tables

**Figure 1 ijms-26-04468-f001:**
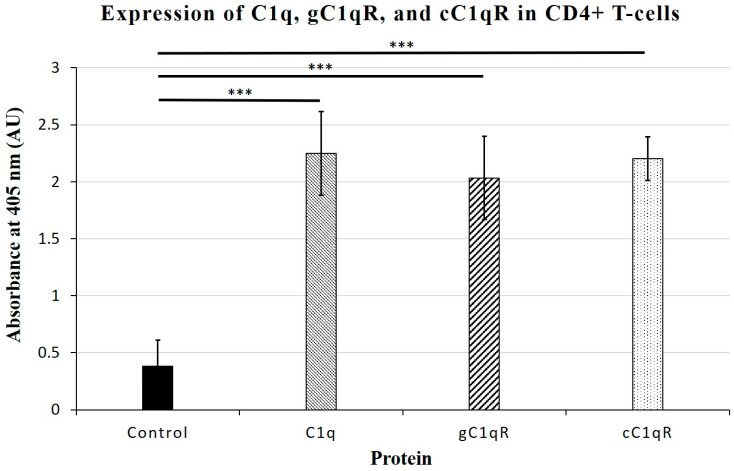
Surface expression of C1q proteins on Molt-4 CD4+ T-cells. Expression of C1q protein and associated receptors, gC1qR and cC1qR, was detected on surface of intact Molt-4 CD4+ T-cells using indirect ELISA run in duplicates. Relative absorbance measured at 405 nm shows C1q with 2.248 ± 0.3689; gC1qR with 2.034 ± 0.3630; and cC1qR with 2.202 ± 0.1905 (n = 4). Negative control wells, which contained no Molt-4 cells, had average absorbance of 0.37975 ± 0.229292208 (n = 4 each). Cell-surface presence was highest for C1q, but all three proteins displayed significant expression. *** *p* < 0.001.

**Figure 2 ijms-26-04468-f002:**
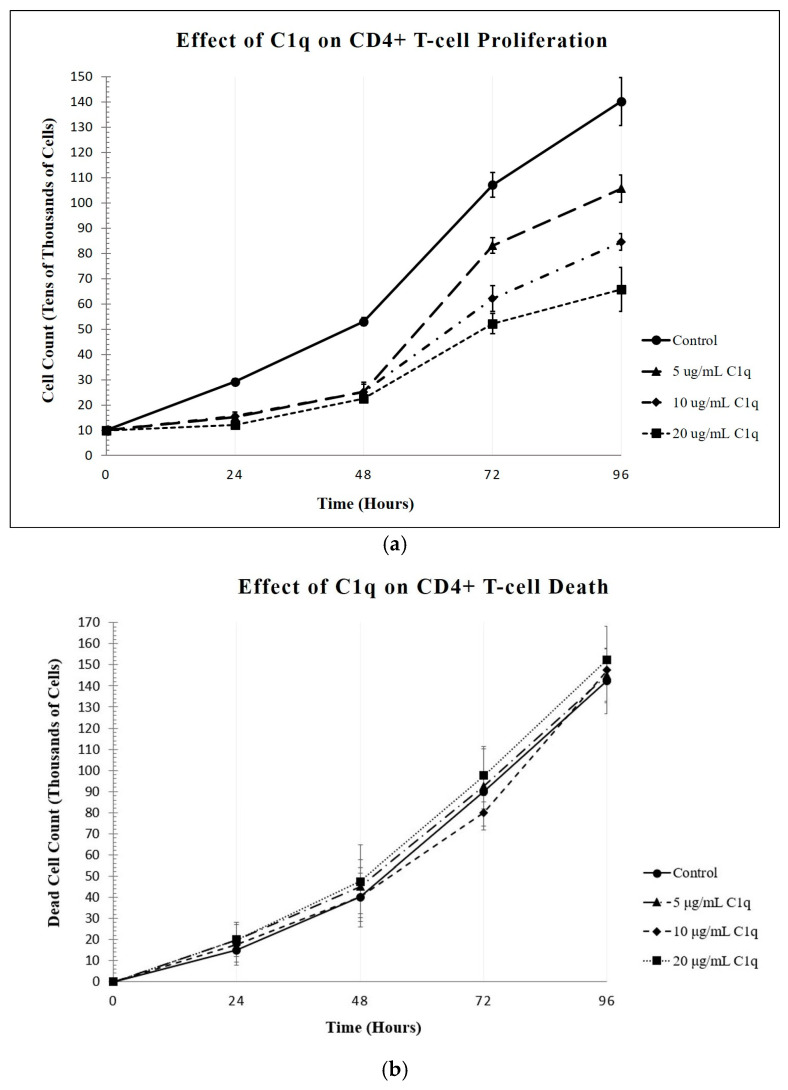
The effects of increasing concentrations of C1q on the proliferation of CD4+ T-cells. Molt-4 cells were cultured in the presence of varying doses of C1q, and then analyzed as described below. (**a**) Effect on live cells: The cells were cultured in the presence or absence of increasing concentrations of C1q, and at 24 h intervals, equal samples were taken from each culture, and the cell viability was determined using a hemocytometer. As shown in the figure, the proliferation of the cells incubated with C1q was diminished in a dose-dependent manner. At 96 h of incubation, the control culture contained 140.200 ± 9.4798 × 10^4^ live cells (n = 3), whereas the culture with 5 μg/mL of C1q contained 105.650 ± 5.4415 × 10^4^ live cells; with 10 μg/mL of C1q, 84.500 ± 3.2787 × 10^4^ live cells; and with 20 μg/mL, 65.733 ± 8.6379 × 10^4^ live cells (n = 3 each). (**b**) Effect on cell viability: A viability assay for each concentration of C1q was also determined at 24 h intervals. At 96 h, the control cell culture contained 148.000 ± 17.6257 × 10^3^ dead cells, whereas those incubated with 5 μg/mL of C1q had 151.250 ± 12.7377 × 10^3^ dead cells, with 10 μg/mL of C1q had 163.500 ± 9.3274 × 10^3^ dead cells, and with 20 μg/mL of C1q contained 157.000 ± 14.7196 × 10^3^ dead cells (n = 3 each).

**Figure 3 ijms-26-04468-f003:**
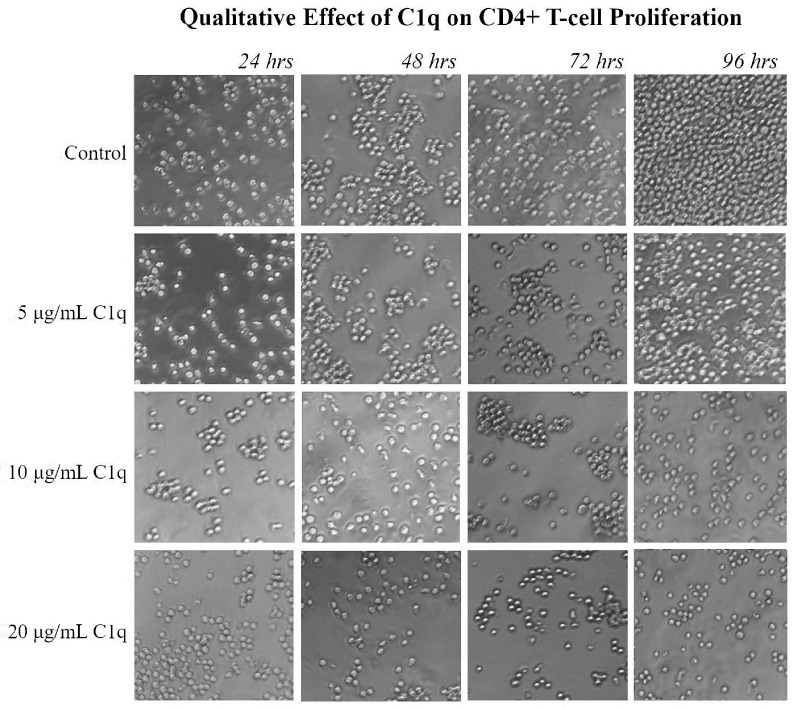
The effects of C1q on the proliferation of CD4+ T-cells. Molt-4 cells were cultured in the presence of varying doses of C1q. The cell cultures were photographed at 24 h intervals under a microscope (40× magnification) for a qualitative translation of the quantitative proliferation assays. The images support the cell counts, as the control sample has a visibly larger number of live cells than the cultures with C1q.

**Figure 4 ijms-26-04468-f004:**
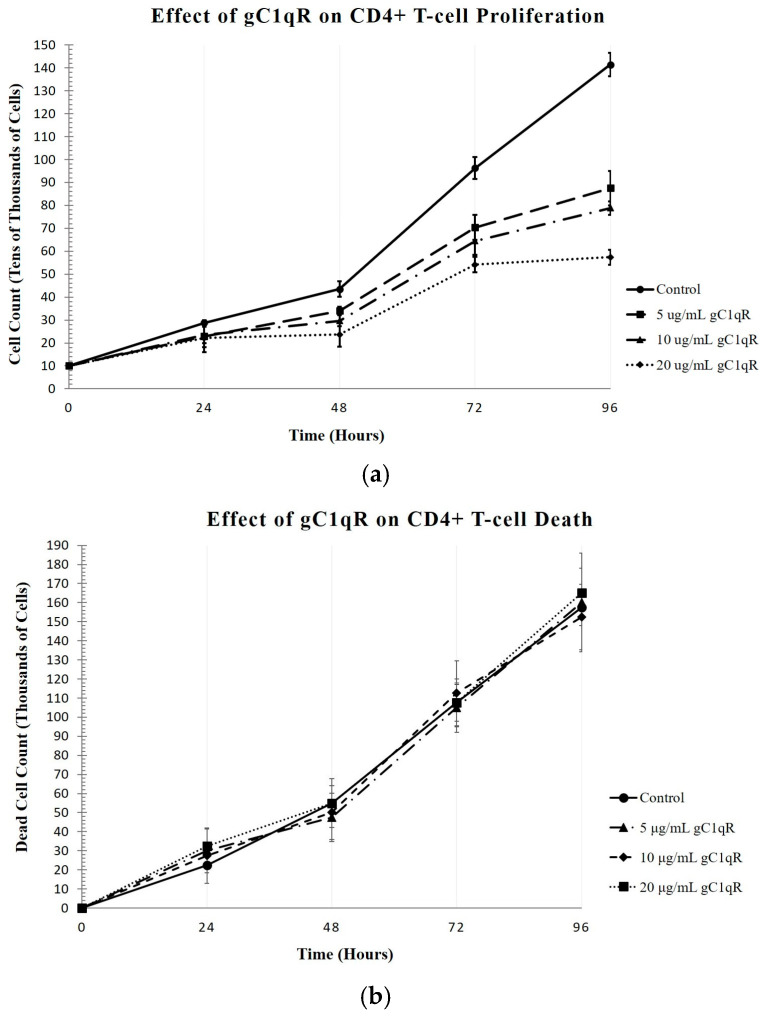
The effects of increasing concentrations of gC1qR on the proliferation of CD4+ T-cells. Molt-4 cells were cultured in the presence of varying doses of gC1qR, and then the cells were counted in the presence of Trypan blue. (**a**) While there was no visible change in the cell count in the control culture, even after 96 h, the cells cultured in the presence of varying doses of gC1qR resulted in diminished cell counts without significant cell death (Figure 4a). At 96 h, the control culture contained 141.400 ± 5.0715 × 10^4^ live cells (n = 3), whereas the culture with 5 μg/mL of gC1qR contained 87.533 ± 7.5002 × 10^4^ live cells, with 10 μg/mL of gC1qR contained 78.867 ± 2.9409 × 10^4^ live cells, and with 20 μg/mL of gC1qR contained 57.333 ± 3.2332 × 10^4^ live cells at 96 h (n = 3 each). (**b**) The effect of gC1qR on the cell viability remained consistent among all samples throughout the 96 h assay at each time interval. At 96 h, the control cell culture contained 157.500 ± 9.5742 × 10^3^ dead cells, whereas with 5 μg/mL of gC1qR, there were 160.000 ± 25.8199 × 10^3^ dead cells; with 10 μg/mL of gC1qR, 152.500 ± 17.0783 × 10^3^ dead cells; and with 20 μg/mL of gC1qR, 165.000 ± 12.9099 × 10^3^ dead cells (n = 3 each).

**Figure 5 ijms-26-04468-f005:**
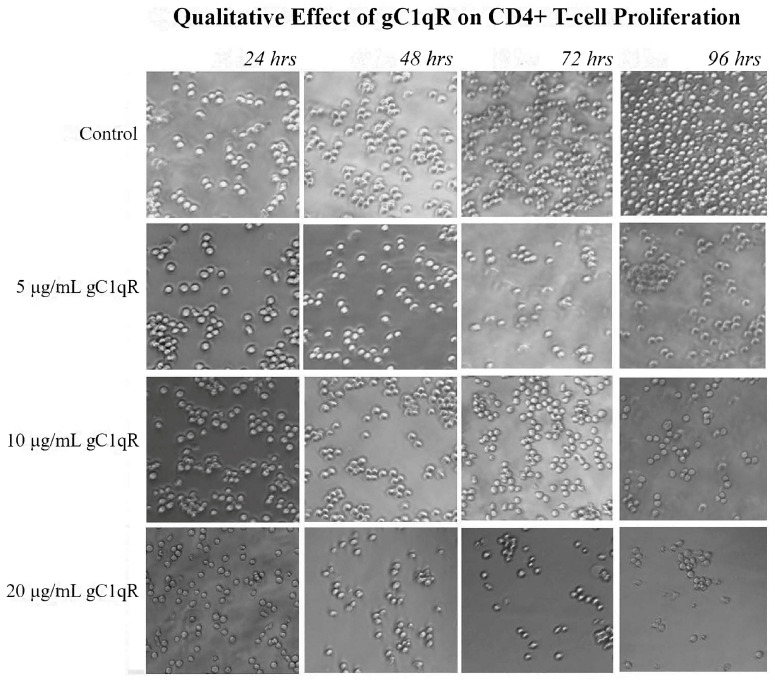
The effect of increasing concentrations of gC1qR on the proliferation of CD4+ T-cells. Molt-4 cells were cultured in the presence of varying doses of gC1qR. The cell cultures were photographed at 24 h intervals under a microscope (40× magnification). The images support the cell counts, in that the control sample has a visibly larger number of live cells after 96 h than the culture with gC1qR.

**Figure 6 ijms-26-04468-f006:**
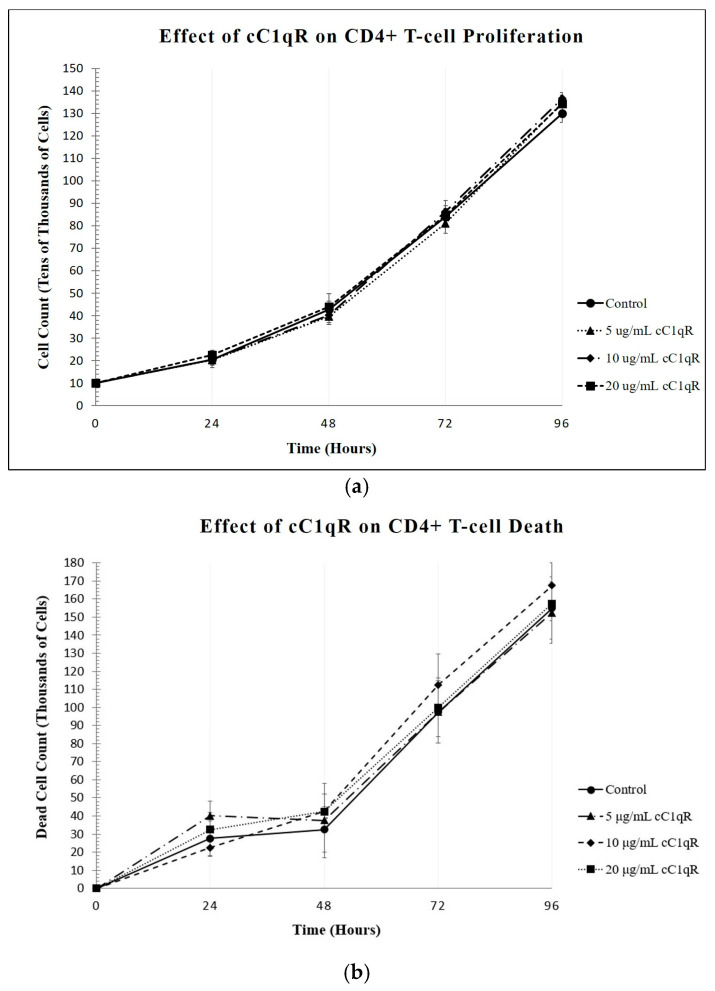
The effects of increasing concentrations of cC1qR on the proliferation of CD4+ T-cells. Molt-4 cells were cultured in the presence of varying doses of cC1qR, and (**a**) the effects on the cells were analyzed as above. There was no significant change in the cell count with an increasing concentration of cC1qR. At 96 h, the control culture contained 130.000 ± 4.0825 × 10^4^ live cells (n = 3). For the cultures with cC1qR (**a**), the culture with 5 μg/mL of cC1qR contained 134.750 ± 3.3040 × 10^4^ live cells, the culture with 10 μg/mL of cC1qR contained 136.500 ± 2.8868 × 10^4^ live cells, and the culture with 20 μg/mL of cC1qR contained 134.05 ± 5.0580 × 10^4^ live cells (n = 3 each). (**b**) shows that the addition of cC1qR did not affect the cell viability after 96 h. At 96 h, the control cell culture contained 156.250 ± 14.4539 × 10^3^ dead cells, the culture with 5 μg/mL of cC1qR contained 164.750 ± 9.8107 × 10^3^ dead cells, the culture with 10 μg/mL of cC1qR contained 153.750 ± 14.8857 × 10^3^ dead cells, and the culture with 20 μg/mL of cC1qR contained 146.500 ± 12.7148 × 10^3^ dead cells (n = 3 each).

**Figure 7 ijms-26-04468-f007:**
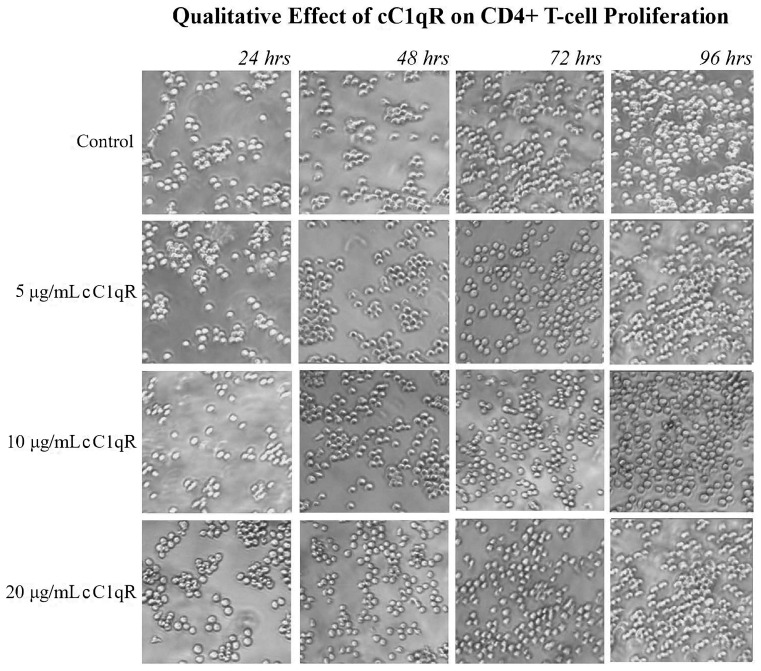
The effect of increasing concentrations of cC1qR on the proliferation of CD4+ T-cells. Molt-4 cells were cultured in the presence of varying doses of cC1qR. The cell cultures were photographed at 24 h intervals under a microscope (40× magnification). The images support the cell counts, as the control sample has a visibly similar number of live cells after 96 h to those cultured with 20 μg/mL of cC1qR.

**Figure 8 ijms-26-04468-f008:**
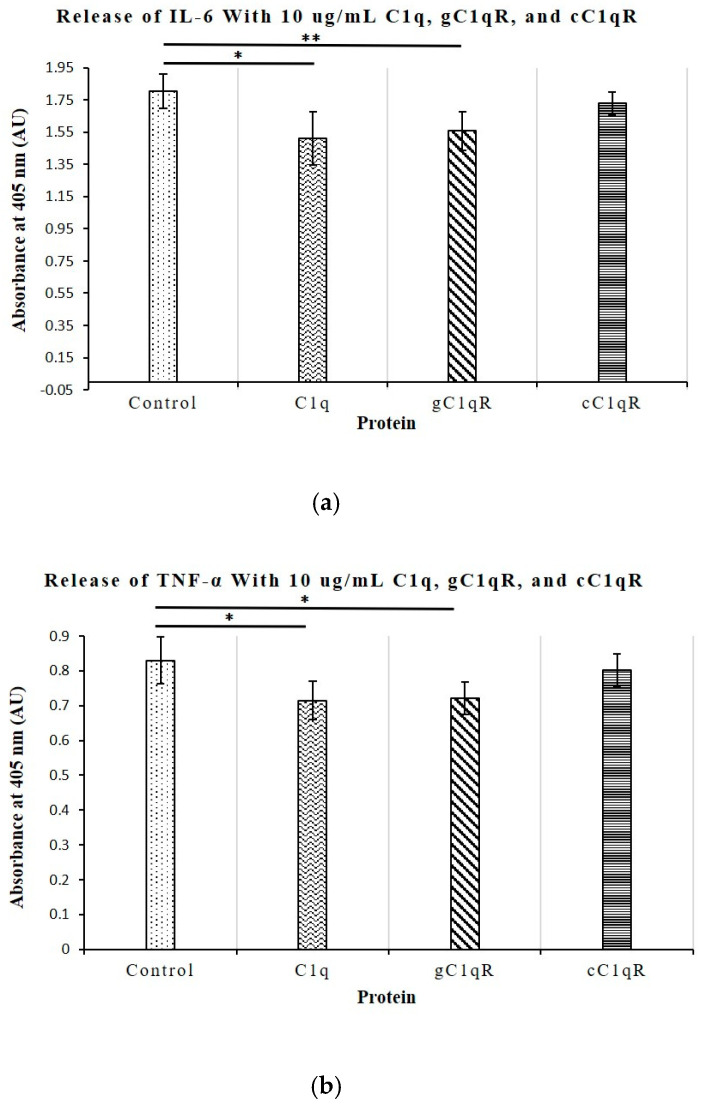
The release of key regulatory pro-inflammatory cytokines from Molt-4 CD4+ T-cells after incubation with C1q, gC1qR, and cC1qR. (**a**) The release of the immunostimulatory cytokine IL-6 was significantly altered when CD4+ T-cells were cultured with 10 μg/mL of C1q and gC1qR for 36 h (n = 4 each); * *p* < 0.05, ** *p* < 0.01, using Student’s *t*-test. (**b**) The release of TFN-α was significantly altered by culturing the CD4+ T-cells with 10 μg/mL of C1q and gC1qR for 36 h (n = 4 each). * *p* < 0.05, calculated using Student’s *t*-test.

**Figure 9 ijms-26-04468-f009:**
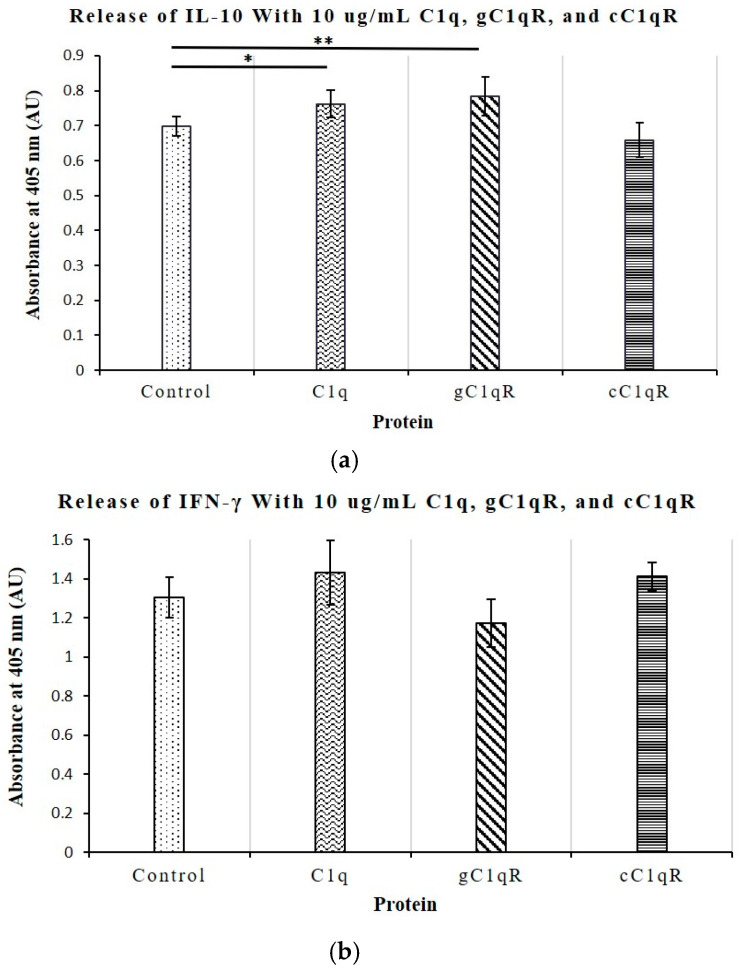
The effects of C1q, gC1qR, and cC1qR on the release of IL-10 from Molt-4 CD4+ T-cells. The release of the immunosuppressant cytokine IL-10 was not seen to be altered when Molt-4 T-cells were incubated with C1q, gC1qR, or cC1qR (*p* > 0.05). The relative absorbances were as follows: control culture—1.444 ± 0.2675; culture with 10 μg/mL of C1q—1.298 ± 0.1425; culture with 10 μg/mL of gC1qR—1.304 ± 0.1545; culture with 10 μg/mL of cC1qR—1.525 ± 0.7949 (n = 4 each); * *p* < 0.05, ** *p* < 0.01. (**a**) The release of the immunostimulatory cytokine INF-γ was not seen to be altered significantly under the influence of C1q, gC1qR, or cC1qR (*p* > 0.05). (**b**) The relative absorbances were as follows: control culture—1.304 ± 0.2786; culture with 10 μg/mL of C1q—1.430 ± 0.7414; culture with 10 μg/mL of gC1qR—1.173 ± 0.2466; culture with 10 μg/mL. of cC1qR—1.411 ± 0.3526 (n = 4 each).

**Table 1 ijms-26-04468-t001:** Effects of C1q and C1q receptors on release of regulatory cytokines from CD4+ T-cells.

Cytokine	Pro-Inflammatory	Anti-Inflammatory	Levels in SLE ^a^	Effect * of C1q	Effect * of gC1qR	Effect * of cC1qR
IL-6	●		↑	↓	↓	N/A
TNF-α	●		↑	↓	↓	N/A
IL-10		●	↑	N/A	N/A	N/A
IFN-γ		●	↑	N/A	N/A	N/A

^a^ The cytokine levels in SLE patient and animal serum samples, as a point of comparison (23). * Effect of indicated protein on level of secretion of respective cytokine from CD4+ T-cells.

## Data Availability

The original contributions presented in this study are included in the article. Further inquiries can be directed to the corresponding author.

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
