# Peer review of "C1q Binds to CD4+ T Cells and Inhibits the Release of Pro-Inflammatory Cytokines: Role in the Pathogenesis of Systemic Lupus Erythematosus"

_ijms, 2025, doi:10.3390/ijms26104468_

Round 1

Reviewer 1 Report

Comments and Suggestions for Authors

Based on my review of the manuscript titled "C1q Binds to CD4+ T Cells and Inhibits the Release of Proinflammatory Cytokines: Role in the Pathogenesis of Systemic Lupus Erythematosus," I provide the following recommendation and suggestions for improvement:

Recommendation: Minor Revision

The manuscript presents an interesting and relevant study on the role of C1q in regulating CD4+ T-cell responses in SLE. The experiments are well-structured, and the findings contribute to the understanding of immune regulation in lupus. However, there are areas that require further clarification and refinement before acceptance.

Areas for Improvement:

  1. Clarity in Rationale and Hypothesis
  • The introduction provides strong background information but could benefit from a more concise and focused hypothesis. The transition from apoptosis to immunoregulation could be made smoother.
  • The role of C1q receptors (gC1qR and cC1qR) in SLE should be explicitly stated earlier in the introduction.
  1. Experimental Considerations
  • The use of Molt-4 cells (a T-cell leukemia cell line) as a model for CD4+ T-cells needs justification. Are primary CD4+ T-cells unavailable? The authors should discuss whether Molt-4 accurately represents physiological CD4+ T-cell function.
  • Were controls included for serum-derived C1q (e.g., heat-inactivated C1q to rule out contaminants)?
  1. Interpretation of Results
  • C1q's Effect on Cytokines: The reduction of IL-6 and TNF-α is described well, but how does this compare to cytokine regulation in actual SLE patient samples?
  • cC1qR’s Lack of Effect: The finding that cC1qR does not influence proliferation should be explored further. Could it still play an indirect regulatory role?
  1. Language and Style
  • The text is generally well-written but contains long and complex sentences. Some repetitive phrasing should be refined.
  • Ensure consistent terminology (e.g., “self-tolerance” vs. “immune tolerance”).

Final Decision:

I recommend a minor revision to improve clarity and provide more discussion on the biological significance of the findings. The revisions do not require additional experiments but should enhance the depth of interpretation.

Comments on the Quality of English Language

Language and Style

  • The text is generally well-written but contains long and complex sentences. Some repetitive phrasing should be refined.
  • Ensure consistent terminology (e.g., “self-tolerance” vs. “immune tolerance”).

Reviewer 2 Report

Comments and Suggestions for Authors

This is an in vitro study to evaluate the involvement of C1q in the onset and progression of systemic lupus erythematosus (SLE) by investigating the effects of C1q and its receptor on the proliferation of CD4-positive T cells. This study demonstrated that the addition of C1q or gC1qR suppressed the proliferation of CD4-positive T cells and inhibited the production of proinflammatory cytokines. Therefore, authors suggest that C1q or C1qR may be involved in suppressing tissue damage in SLE.

The presented study was well performed, and the manuscript is described in a reasonable manner. This study was performed with Molt-4 CD4+ T cells, but are CD4+ T cells from SLE patients similar to Molt-4 CD4+ T cells? For example, is the expression of C1q and its receptors on CD4+ T cells from SLE patients similar to that of Molt-4 CD4+ T cells? In addition, in experiments in which CD4-positive T cell proliferation was examined by adding C1q or C1qR, was there any change in the expression of C1q or C1qR on the cell surface of CD4-positive T cells after 96 hours of culture?

Moreover, there were some points of concern:

  1. Please provide the statistical method used to compare groups to the statistical analysis section.
  2. Would an empty well be appropriate as the control in Figure 1? Wouldn't a well containing only cells be better?
  3. Please correct the cell number in the footnote of Figure 4 and 6 (e.g. 104 cells, not 104 cells).

Round 2

Reviewer 2 Report

Comments and Suggestions for Authors

This is an in vitro study to evaluate the involvement of C1q in the onset and progression of systemic lupus erythematosus (SLE) by investigating the effects of C1q and its receptor on the proliferation of CD4-positive T cells. This study demonstrated that the addition of C1q or gC1qR suppressed the proliferation of CD4-positive T cells and inhibited the production of proinflammatory cytokines. Therefore, authors suggest that C1q or C1qR may be involved in suppressing tissue damage in SLE.

The presented study was well performed, and the revised manuscript is described in a reasonable manner. Authors had also responded to my all comments.